# A Novel Mechanism of bta-miR-210 in Bovine Early Intramuscular Adipogenesis

**DOI:** 10.3390/genes11060601

**Published:** 2020-05-29

**Authors:** Ling Ren, Qian Li, Xin Hu, Qiyuan Yang, Min Du, Yishen Xing, Yahui Wang, Junya Li, Lupei Zhang

**Affiliations:** 1Key Laboratory of Animal Genetics Breeding and Reproduction, Ministry of Agriculture and Rural Affairs, Institute of Animal Sciences, Chinese Academy of Agricultural Sciences, Beijing 100193, China; renling5454@163.com (L.R.); lq798711247@163.com (Q.L.); huxin19890803@163.com (X.H.); yishen_xing@163.com (Y.X.); wang1434243198@163.com (Y.W.); lijunya@caas.cn (J.L.); 2Molecular and Cellular Biology, Gembloux Agro-Bio Tech, University of Liège, 5030 Gembloux, Belgium; 3Department of Molecular, Cell and Cancer Biology, University of Massachusetts Medical School, Worcester, MA 01655, USA; qiyuan.yang@umassmed.edu; 4Washington Center for Muscle Biology and Department of Animal Sciences, Washington State University, Pullman, WA 99164, USA; min.du@wsu.edu

**Keywords:** bta-miR-210, bovine, progenitor cells, intramuscular adipogenesis, *WISP2*

## Abstract

Intramuscular fat (IMF) is one of the major factors determining beef quality. IMF formation is influenced by multiple conditions including genetic background, age and nutrition. In our previous investigation, bta-miR-210 was found to be increased during adipogenesis using miRNA-seq. In this study, we validated the upregulation of bta-miR-210 in platelet-derived growth factor receptor α positive (PDGFRα^+^) progenitor cells during adipogenic differentiation in vitro. To investigate its role in adipogenesis, bta-miR-210 mimics were introduced into progenitor cells, which resulted in enhanced intracellular lipid accumulation. Accordingly, the expression of adipocyte-specific genes significantly increased in the bta-miR-210 mimic group compared to that in the negative control group (*p* < 0.01). Dual-luciferase reporter assays revealed that *WISP2* is a target of bta-miR-210. *WISP2* knockdown enhanced adipogenesis. In conclusion, bta-miR-210 positively regulates the adipogenesis of PDGFRα^+^ cells derived from bovine fetal muscle by targeting *WISP2*.

## 1. Introduction

Intramuscular fat (IMF) is the term describing the adipose tissue deposited in the muscle and is associated with meat flavor and juiciness [1,2]. IMF refers to the adipose tissue located within muscles and is different from intermuscular fat that accounts for the fat between different muscles. IMF appears as adipose depots with adipocytes between the bundles of muscle fibers and embedded in the connective tissue matrix near a blood capillary system [3]. IMF accumulation is associated with both increased adipocyte size and numbers [4,5].

Skeletal muscle development mainly involves myogenesis but also includes adipogenesis and fibrogenesis. The fetal period is important for muscle development because there is no significant fiber number increase after birth [6]. Myogenic cells are developed from mesenchymal stem cells (MSCs) in the mesoderm. During primordial muscle development, the vast majority of MSCs commit to myogenesis [7]. Fetal and neonatal periods are crucial for bovine (*Bos taurus*) IMF development [8]. A small portion of MSCs in fetal skeletal muscle commit to pre-adipocytes, which differentiate into adipocytes to form the accumulation sites in the subsequent development [9,10]. The early adipogenic commitment initiates from the end of the first trimester for beef cattle, and most adipocytes are formed during late gestation to the neonatal stage [8]. Enhancing the number of MSCs committed to pre-adipocytes will improve IMF deposition.

Adipogenesis is a complex process that is controlled by several key transcription factors. The most essential regulator in the adipogenic differentiation of precursor cells is peroxisome proliferator-activated receptor γ (*PPARγ*) [11]. The transcriptional activator zinc finger protein 423 (*ZFP423*) functions as a PPARγ activator and plays crucial roles in preadipocyte commitment. A decreased expression of *ZFP423* in 3T3-L1 cells inhibits the expression of preadipocyte *PPARγ* [12].

MicroRNAs (miRNAs) belongs to non-coding RNAs and are characterized by lengths of approximately 20 to 22 nucleotides. miRNAs are broadly expressed in almost all eukaryotes and mainly involved in gene post-transcriptional regulation [13], which regulates various cellular processes [14,15,16]. Adipogenesis is also regulated by miRNAs [17]. Previous studies have shown that mmu-miR-210 is remarkably up-regulated during 3T3-L1 cell adipogenesis [18,19], suggesting that miR-210 may regulate adipogenic differentiation. Additionally, miR-210 improves adipogenesis by inhibiting the Wnt signaling pathway in the 3T3-L1 cell line [20]. Using an in vitro intramuscular adipogenesis model and miRNA-seq technique, we found that *Bos taurus* miR-210 (bta-miR-210) was the most significantly upregulated miRNA during the adipogenic differentiation of platelet-derived growth factor receptor α positive (PDGFRα^+^) progenitor cells [21]. However, the mechanism of bta-miR-210 in regulating bovine intramuscular adipogenesis has not been determined.

Due to the importance of fetal stage in IMF development, in the present study, we attempted to explore the function of bta-miR-210 in the adipogenesis of bovine fetal muscle-derived progenitors. We found that bta-miR-210 expression increased during the adipogenic differentiation of PDGFRα^+^ cells and that the overexpression of bta-miR-210 led to an enhanced expression of adipocyte-specific markers. We further found that bta-miR-210 promoted PDGFRα^+^ cell adipogenesis through Wnt signaling pathway inhibition via targeting *WISP2*, which was previously reported as an inhibitor of the transcriptional activator ZFP423 by forming a cytosolic complex with ZFP423 to prevent it entering the nucleus and activating *PPARγ* [22].

## 2. Materials and Methods

### 2.1. Animal

Animal experiments were performed according to the requirements of the Administration of Affairs Concerning Experimental Animals (Ministry of Science and Technology, China, 2004). The animal experiments were endorsed by the Animal Ethics Committee of the Institute of Animal Science, Chinese Academy of Agricultural Sciences. Gravid cows were raised in Jingxinxufa Agriculture Co., Ltd. (Weichang, China). All efforts were made to minimize both cows’ and fetuses’ suffering. In the current study, three simmental fetuses at 123,136 and 146 days of pregnancy were used. The cows were stunned and bled out. The uterus was removed from slaughter cow, and the fetus was left undisturbed in utero for at least 30 min. The uterus was transferred immediately to a culture room, and then the fetus was removed from the uterus. Bovine tissues including the longissimus dorsi, leg muscle, heart, liver, spleen, lung, kidney, stomach and small intestine were sampled from fetuses. Samples were instantly rinsed in phosphate-buffered saline (PBS) and snap-frozen and stored in liquid nitrogen.

### 2.2. Cell Isolation and Culture

Fetal longissimus dorsi muscle was used for the isolation of PDGFRα^+^ cells and cultured as previously described [21]. Briefly, the fetus was transferred to the lab within 3 hours. The longissimus dorsi was cut into pieces and then digested using 0.1% type IV collagenase (Sigma, St. Louis, MO, USA) for 1 hour. The reaction was stopped with growth medium comprising low-glucose Dulbecco’s Modified Eagle Medium (DMEM; Gibco, Grand Island, NY, USA) containing 1 mM Sodium Pyruvate, 4 mM L-Glutamine, 10% fetal bovine serum (FBS; Gibco) and 100 U/mL of penicillin-streptomycin (Gibco). The suspension was filtered through a 40 µm mesh, and the pellet was collected and resuspended in PBS comprising 2 mM EDTA (Solarbio, Beijing, China) and 0.5% Fraction V (Beyotime, Shanghai, China). The cells were incubated with anti-PDGFRα antibodies (Cell Signaling Technology, Danvers, MA, USA) at 4 °C for half an hour. After rinsing, the cells were reacted with Anti-Rabbit IgG MicroBeads (Miltenyi Biotec, Bergisch Gladbach, Germany) at 4 °C for 20 min. Subsequently, the cells were centrifuged and resuspended. The PDGFRα^+^ cells were isolated using MACS columns (Miltenyi Biotec, Bergisch Gladbach, Germany) and a magnetic MiniMACS Separator (Miltenyi Biotec, Bergisch Gladbach, Germany). The PDGFRα^+^ cells were raised in growth medium at 37 °C in a humidified atmosphere containing 5% CO_2_. The cells isolated from each fetus were maintained separately. After reaching 75% confluence, the cells were detached with 0.25% trypsin (25200056, Gibco, Grand Island, NY, USA) and subcultured into sterile 12-well plates (Corning, 3.5 cm^2^/well, Corning, NY, USA) at a density of 10,000 cells/cm^2^ for subsequent studies or frozen in liquid nitrogen. Only Passage 2 to 5 primary cells were used in this investigation.

The human embryonic kidney (HEK) 293T cell line was obtained from the China Center for Type Culture Collection and raised in high-glucose DMEM containing 1 mM Sodium Pyruvate, 4 mM L-Glutamine, 10% fetal bovine serum (FBS; Gibco) and 100 U/mL of penicillin-streptomycin (Gibco) at 37 °C in a humidified atmosphere containing 5% CO_2_.

### 2.3. Adipogenic Differentiation

At full confluence, PDGFRα^+^ cells were exposed to adipogenic medium consisting of low-glucose DMEM supplemented with 10% FBS (Gibco), 1 mM dexamethasone (D1756, Sigma-Aldrich, St. Louis, MO, USA), 0.5 mM isobutyl-methylxanthine (I5879, Sigma-Aldrich, St. Louis, MO, USA), 3 μg/mL of insulin (I5500, Sigma-Aldrich, St. Louis, MO, USA), 1 mM rosiglitazone (R2408, Sigma-Aldrich, St. Louis, MO, USA) and 100 U/mL of penicillin-streptomycin (Gibco). After two days, the cells were switched to maintenance medium consisting of low-glucose DMEM (Gibco) supplemented with 3 μg/mL of insulin, 1 mM rosiglitazone, and 100 U/mL of penicillin-streptomycin (Gibco).

### 2.4. Transfection

Bta-miR-210 mimics, mimic negative control (NC), *WISP2* siRNA and siRNA NC were purchased from RiboBio Co. Ltd. (Guangzhou, China), and the sequences are as follows: si-WISP2-001: 5′-CCCGAGUGUCCAAUCAGAA-3′, si-WISP2-002: 5′-CCUGAAGGACAAGCGUAUU-3′, si-WISP2-003: 5′-GGACCCUUAAAUGUCCCUU-3′, and bta-miR-210 mimics: 5′-ACUGUGCGUGUGACAGCGGCUGA-3′. 

After reaching 70–80% confluence, the PDGFRα⁺ cells were detached, counted, and seeded into 12-well plates (Corning) at a density of 10,000 cells/cm^2^. Lipofectamine RNAiMAX (13778-075, Invitrogen Life Technologies, Carlsbad, CA, USA) reagent and mimics/siRNAs were diluted in serum-free DMEM (Gibco) separately. The diluted RNA and the diluted Lipofectamine RNAiMAX were combined and incubated for 20 min. The RNA–RNAiMAX complexes were then added to each well. The final concentration was 50 nM. 

For the luciferase reporter assay, HEK293T cells were detached, counted, and seeded into 96-well plates (Corning) at a density of 10,000 cells/cm^2^ after reaching 70–80% confluence. Lipofectamine 2000 (11668027, Invitrogen Life Technologies, Carlsbad, CA, USA) reagent was used to induce miRNA mimic and reporter vector uptake by the HEK293T cells. Lipofectamine 2000 and double-stranded RNA (dsRNA) mimics/plasmid DNA were diluted in serum-free DMEM (Gibco) separately. The diluted mimics and plasmids and the diluted Lipofectamine 2000 were combined and incubated for 20 min. The RNA–lipid complexes were then added to the HEK293T cells. The final concentrations of the mimics and plasmid were 50 nM and 1 μg/μL, respectively.

### 2.5. Quantitative Real-Time PCR

Fetal tissues were homogenized with a TissueLyser (Qiagen, Hilden, Germany) in TRIzol (15596026, Invitrogen, Carlsbad, CA, USA). The cells were lysed with TRIzol. Chloroform was mixed with the lysates vigorously. After centrifuging, the aqueous phase was transferred into a new tube. Isopropanol was used to precipitate the RNA from the aqueous phase. After washing the RNA pellets with 75% ethanol, the resulting RNA pellets were then resuspended in RNase-free water. The RNA concentrations and purity were analyzed using a NanoPhotometer N50 (Implen, Munich, Germany). Reverse transcription was executed using the PrimeScript^TM^ RT Master Mix (RR036A, TaKaRa, Kusatsu, Japan) for mRNA and the miRcute Plus miRNA First-Strand cDNA Kit (KR211, Tiangen, Beijing, China) for miRNA. Quantitation was analyzed using the ABI QuantStudio 7 Flex system. The primers for bta-miR-210 and the *PPARγ*, *ZNF423*, CCAAT enhancer binding protein alpha (*C/EBPα*), fatty acid binding protein 4 (*FABP4*), *WISP2* and *18S* genes used in qRT-PCR are listed in Table 1.

### 2.6. Oil Red O Staining

On Day 9 of adipogenic differentiation, the medium was discarded and the cells were rinsed once in PBS and fixed by incubating with 4% paraformaldehyde for fifteen minutes. The cells were then incubated with 0.5% Oil Red O (ORO, O0625, Sigma-Aldrich, St. Louis, MO, USA) for 15 min to assess intracellular lipid accumulation. Next, we eluted the ORO from lipid droplets with isopropanol and quantified it with a spectrophotometer (Bio-Rad, Hercules, CA, USA) at a 490 nm wavelength [23].

### 2.7. Western Blotting

Protein samples were lysed with lysis buffer (P0013, Beyotime, Shanghai, China) containing 20 mM Tris (pH 7.5), 150 mM NaCl, 1% Triton X-100, 50 mM NaF and 1 mM Na_3_VO_4_. The total protein concentration was analyzed with the Enhanced BCA Protein Assay Kit (P0010S, Beyotime, Shanghai, China). Equal amounts of protein (20 μg) from the cell lysates were loaded for each lane, separated and transferred to a polyvinylidene difluoride (PVDF) membrane (Millipore, Bedford, MA, USA). Because of the close sizes of β-tubulin and PPARγ, the same samples were separated for two membranes. The membranes were blocked with 5% (*w/v*) nonfat dried milk at room temperature for one hour, and then incubated the target proteins with following primary antibodies against ZNF423 (sc-10486, Santa Cruz Biotechnology, Santa Cruz, CA, USA), PPARγ (sc-6284, Santa Cruz Biotechnology, Santa Cruz, CA, USA), FABP4 (sc-18661, Santa Cruz Biotechnology, Santa Cruz, CA, USA), WISP2 (sc-514070, Santa Cruz Biotechnology, Santa Cruz, CA, USA) and β-tubulin (10094-1-AP, Proteintech, Chicago, IL, USA), with 1:1000 dilution at 4 °C overnight. The membranes were rinsed three times and reacted with HRP-conjugated donkey anti-goat IgG (A0181, Beyotime, Shanghai, China) or HRP-conjugated goat anti-mouse IgG (A0216, Beyotime, Shanghai, China) for 1 hour. The signals were activated using the ECL Western blot detection reagent (RPN2106, GE Healthcare Biosciences, Pittsburgh, PA, USA).

### 2.8. Bioinformatics Analysis

The TargetScan website (http://www.targetscan.org/vert_72/) was used to predict the target genes of the miRNA. The predicted target genes with a context ++ score lower than −0.3 were selected for subsequent GO (gene ontology) and Kyoto Encyclopedia of Genes and Genomes (KEGG) analysis. The GO term analysis and the KEGG pathway analysis were performed with DAVID Bioinformatics Resources (https://david.ncifcrf.gov/), respectively. *p* < 0.05 was set as the cut-off criterion.

### 2.9. DNA Constructs and Luciferase Reporter Assays

The segment containing potential binding sites of miR-210 was amplified from the *WISP2* 3’ untranslated region (UTR) and recombined at the 3′ end of the SV40 promoter-driven *Renilla* luciferase gene in the psiCHECK-2 vector (C8021, Promega, Madison, WI, USA). The other luciferase gene in psiCHECK-2, the firefly luciferase gene, was driven by the HSV-TK promoter and independently transcribed and used as an internal control. The Fast Site-Directed Mutagenesis Kit (KM101, Tiangen Biotech, Beijing, China) was used to induce site-directed mutagenesis. Table 2 shows the primers used in plasmid recombination and mutagenesis. HEK293T cells were co-transfected with bta-miR-210 mimics or a scrambled miRNA control and psiCHECK-*WISP2* or its mutant plasmids. Forty-eight hours after transfection, the luciferase activity was analyzed with the Dual-Luciferase Reporter System (Promega Corporation, Madison, WI, USA). The growth medium was removed, and the cells were washed with PBS. The cells were lysed with passive lysis buffer. Ten microliters of lysate per sample were transferred to a 96-well plate. The firefly luciferase activity was measured with Infinite 200 PRO (TECAN, Switzerland) after adding Luciferase Assay Buffer II. Then, the *Renilla* luciferase activity was measured after adding Stop & Glo Reagent.

### 2.10. Statistical Analysis

The relative mRNA expression level was calculated with the 2^−ΔΔCt^ algorithm by normalizing to the expression of 18S. The relative average grey levels of the Western blots were analyzed using the Image J software. The protein level was normalized to β-tubulin. The fluorescence values of Renilla were normalized to those of firefly in the dual-luciferase assay. The results of the qPCR, ORO quantification, grayscale intensity determination, and luciferase reporter assays from the cell culture experiments are presented as independent biological replicates, while the tissue qPCR analyses were analyzed using three individual biological replicates. At least three biological replicates were used, and the data were presented as mean ± S.E.M. The data were tested with a variance homogeneity test and analyzed using Student’s t-test with the GraphPad Prism v6.0 (Graphpad Software Inc., La Jolla, CA, USA) for qRT-PCR, ORO staining and luciferase reporter assays. *p* < 0.05 was considered statistically significant, and *p* < 0.01, to show an extremely significant difference.

## 3. Results

### 3.1. miR-210 Expression Increases during Bovine Intramuscular Adipogenesis

PDGFRα+ cells were isolated from bovine fetal skeletal muscle and induced adipogenesis after reaching 100% confluence. These cells exhibited a spindle shape during proliferation (Figure 1a). After adipogenic induction, the cells changed into an oblate shape, and lipid microdroplets could be observed on Day 9 (Figure 1a). ORO staining showed the existence of lipid microdroplets (Figure 1a). Consistently, the mRNA expression levels of *PDGFRA* and the adipocyte-specific genes *PPARγ*, *C/EBPα* and *FABP4* increased significantly (Figure 1b).

Firstly, we analyzed the bta-miR-210 expression profile in fetal tissues and differentiating progenitor cells. miR-210 expression in nine tissues was analyzed, including the longissimus dorsi, leg muscle, heart, liver, spleen, lung, kidney, stomach and small intestine. The results showed that miR-210 was moderately expressed in skeletal muscle tissues (Figure 2a). Moreover, miR-210 expression increased by about 5-fold during adipogenic differentiation (Figure 2b). These data suggested that miR-210 is involved in the differentiation of intramuscular adipocytes.

### 3.2. miR-210 Enhances the Adipogenic Differentiation of PDGFRα^+^ Cells

For better understanding the effects of miR-210 on adipogenic differentiation, bta-miR-210 mimics or non-targeting NC dsRNAs were transfected into PDGFRα+ cells and adipogenic differentiation was induced. The cells were analyzed on Day 9. The ORO staining results showed increased intracellular lipid accumulation in the miR-210 mimics group compared to that in the NC. The OD value of eluted ORO stain in the miR-210 mimics group was considered highly significantly different to that of the NC group (*p* < 0.01) (Figure 3a). After transfection, the miR-210 amount was significantly increased (*p* < 0.01) (Figure 3b). In response to exogenous miR-210, the mRNA expression of *ZNF423*, *PPARγ,* and *C/EBPα* on Day 2 was significantly enhanced (*p* < 0.01), and the expression of PPARγ, C/EBPα, and FABP4 on Day 9 was significantly upregulated (*p* < 0.01) (Figure 3c). Consistently, the protein expression of PPARγ and FABP4 significantly increased after miR-210 transfection (Figure 3d), showing that miR-210 enhances adipogenesis.

### 3.3. miR-210 Regulates Adipogenic Differentiation of PDGFRα^+^ Cells by Inhibiting WISP2

Based on prediction using the TargetScan software, target genes with a context ++ score lower than −0.3 were selected for GO and KEGG analysis. We further performed GO term analysis for miR-210 target genes, from which we found that miR-210 may participate in the regulation of the JNK cascade and MAP kinase activity as well as brown fat cell differentiation (Figure 4a). Moreover, KEGG pathway analysis revealed that miR-210 may regulate biological processes through the PI3K-Akt, FoxO and steroid hormone biosynthesis signaling pathways (Figure 4b). Among the miR-210 target genes, we focused on the *WISP2* gene, which is not assigned to any pathway but demonstrated as a regulator of adipogenesis. There were two potential binding sites located in Positions 240–247 and 347–353 of the *WISP2* 3′-UTR (Figure 5a). Both sites are not conserved among mammals (Figure 5b). To test whether *WISP2* was regulated by miR-210, luciferase reporter assays were performed. Luciferase reporter vectors containing the wild-type 3′-UTR of *WISP2* (*WISP2* WT), mutant at Binding Site 1 (*WISP2* Mut1), mutant at Binding Site 2 (*WISP2* Mut2) or mutant at both sites (*WISP2* Mut3) were constructed and separately transfected along with miR-210 into 293T cells. Compared with the mimic NC, the luciferase activity for the *WISP2* WT group transfected with bta-miR-210 mimics substantially declined (*p* < 0.01); that in the *WISP2* Mut1 and Mut2 groups also decreased (*p* < 0.05), while *WISP2* Mut3 showed no difference (Figure 5c). Consistently, transfection with bta-miR-210 mimics led to significant decline in WISP2 protein content (Figure 5d). These data demonstrated that miR-210 targets the 3′-UTR of *WISP2*.

### 3.4. Silencing WISP2 Improves the Adipogenic Differentiation of PDGFRα⁺ Cells

Corresponding to the isoforms of *WISP2* transcript, three siRNAs were designed. The interference efficiency of siRNAs was verified in PDGFRα⁺ cells. si-*WISP2*-001 demonstrated the highest interference efficiency (Figure 6a), which was used in the subsequent experiment.

PDGFRα⁺ cells transfected with *WISP2* siRNAs or non-targeting siRNA NC were induced for adipogenic differentiation. On Day 9, ORO staining was performed, and greater intracellular lipid accumulation was found for the siRNA group (Figure 6b) Furthermore, the expression of the adipocyte-specific marker genes *ZNF423*, *PPARγ*, *C/EBPα* and *FABP4* was upregulated significantly (*p* < 0.05 for *ZNF423*, *p* < 0.01 for the other genes) (Figure 6c). In agreement, silencing *WISP2* significantly enhanced the protein content of the adipocyte-specific proteins PPARγ and FABP4 (Figure 6d). In summary, these findings showed that miR-210 promotes the adipogenesis of PDGFRα⁺ cells by inhibiting *WISP2* expression.

## 4. Discussion

Skeletal muscle is a complex organ and composed of multiple types of stem cells [2]. Previous studies demonstrated that PDGFRα^+^ muscle-derived stem cells have high adipogenic potential [24]. PDGFRα is a receptor that binds to platelet-derived growth factors (PDGFs), triggering cellular growth and differentiation [25,26]. PDGFRα regulates the equilibrium of stromal and adipogenic cells during adipose tissue development [27]. PDGFRα^+^ progenitor cells display the ability to differentiate into both adipocytes and fibroblasts [24]. Location near blood vessels and capillaries facilitates the ability of PDGFRα^+^ progenitors to sense metabolic changes or adipogenic signals and contributes to adipogenesis and triglyceride storage. High-fat feeding, which is similar to beef cattle fattening, greatly increases the recruitment of white adipocytes from PDGFRα^+^ progenitors [28]. Moreover, the PDGFRα content in Angus skeletal muscle is positively associated with IMF when compared with Nellore [29]. Thus, in this study, we isolated bovine fetal muscle-derived PDGFRα⁺ cells and induced adipogenic differentiation to study the function of miR-210 in intramuscular adipogenesis. Our data show that bovine fetal muscle-derived PDGFRα⁺ cells are suitable for studying bovine intramuscular adipogenesis. The method used in our study provides a valuable model for investigating bovine intramuscular adipogenesis, especially in the early developmental stage without visible IMF.

The roles of miRNAs in regulating economic traits have been widely investigated in pigs, beef cattle and dairy cattle [21,30,31,32]. In our previous investigation, we found that miR-210 was significantly enhanced during intramuscular adipogenesis [21]. miR-210 has been demonstrated to be involved in various biological processes. We first identified that bta-miR-210 was expressed in multiple tissues. The low variance of the bta-miR-210 expression level among three fetuses also implied that the fetuses used in this study were consistent. miRNA-210 regulates angiogenesis and apoptosis by targeting the ephrin A3 and protein tyrosine phosphatase non-receptor type 1 genes, respectively [33]. miR-210 also participates in testis development via regulating nuclear receptor subfamily 1, Group D, Member 2 (NR1D2) [34]. In glioblastoma multiforme (GBM) cells, increased miR-210 expression could induce proliferation and inhibit apoptosis by inhibiting the regulator of differentiation 1, which is associated with GBM function [35]. For endothelial cells, miR-210 controls angiogenesis by modulating ephrin-A3 (Efna3), a factor functioning in the development of the cardiovascular system [36]. As a hypoxia-inducible factor 1 (HIF-1) inducible miRNA, miR-210 inhibits glycerol-3-phosphate dehydrogenase 1-like, which increases HIF-1 stability [37]. The iron-sulfur cluster scaffold homolog and cytochrome c oxidase assembly protein genes, regulators of mitochondrial and tricarboxylic acid cycle function, have been reported as targets of miR-210 [38]. During the differentiation of satellite cells, the expression of miR-210 was upregulated, indicating the potential involvement of miR-210 in trout myogenesis [39]. Furthermore, the miR-210 expression level also increased during myogenic differentiation with a Hif1a-dependent mechanism in normoxia. Under metabolic stress conditions, the increased level of miR-210 seems to be of benefit to increase myofiber survival [40]. Because miRNAs can be secreted, the miR-210 expression in myocytes may also affect intramuscular fat development.

In previous studies, miR-210 has been shown to positively regulate adipogenesis by targeting transcription factor 7 like 2 (*TCF7L2*) [20]. TCF7L2 is a transcription factor that can trigger a downstream reaction of the Wnt signaling pathway. Activating the canonical Wnt pathway causes an increased expression of β-catenin, which subsequently translocates into the nuclei and binds to transcription factors of the TCF/LEF family [41]. The Wnt signaling pathway regulates MSC maintenance, proliferation, fate decision and adipogenesis [42,43,44]. In the 3T3-L1 cell line, the overexpression of *Wnt1* activates the canonical pathway, leading to repressed adipogenesis [45]. By contrast, the interruption of Wnt signaling led to spontaneous adipocyte differentiation [45,46]. However, the miR-210 binding site in the mouse *TCF7L2* 3′-UTR is poorly conserved, and bovine *TCF7L2* does not contain a potential target of bta-miR-210. Hence, we speculated that there might be a different mechanism by which miR-210 regulates bovine intramuscular adipogenesis. Using the TargetScan website, *WISP2* was predicted as a potential target of bta-miR-210. WISP2 is an activator of the canonical Wnt pathway [47], and previous reports indicated that WISP2 is involved in adipogenesis. WISP2 partners with ZFP423, a key regulator of preadipocyte commitment, to prevent its entry into nuclei, thereby inhibiting PPARγ activation and suppressing the adipogenic differentiation of 3T3-L1 adipose cells [12,22,48,49]. In alignments, miRNA-450a-5p is a positive regulator of adipogenesis in rat preadipocytes by inhibiting *WISP2* expression [50]. We have demonstrated a novel regulatory mechanism of miR-210 in bovine adipogenesis, which is distinct from that in mice (Figure 7).

## 5. Conclusions

Taken together, we established that miR-210 played a positive role in regulating the adipogenesis of bovine fetal muscle-derived PDGFRα⁺ cells by targeting *WISP2* at sites poorly conserved among cows, humans, pigs, mice and sheep. Because the fetal stage is critical for intramuscular adipocyte formation in beef cattle, our data suggest that miR-210 is a potential molecular target for enhancing intramuscular adipogenesis and, thus, the IMF content of beef cattle.

## Figures and Tables

**Figure 1 genes-11-00601-f001:**
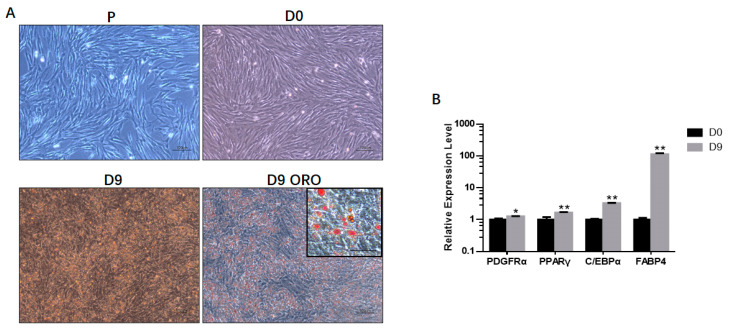
Adipogenic differentiation of bovine fetal skeletal muscle-derived PDGFRα⁺ cells. (**A**) Shown are proliferating PDGFRα+ cells (P) on Day 0 (D0) and Day 9 (D9) following adipogenic differentiation. Differentiated cells on Day 9 were stained with Oil Red O (D9 ORO). Scale bar = 100 μm, inset scale bar = 50 μm. (**B**) Relative gene expression of adipocyte-specific genes on Day 0 (D0) and Day 9 (D9). All the measurements shown are the means ± SEM of four biological replicates from independent cell experiments. * *p* < 0.05; ** *p* < 0.01.

**Figure 2 genes-11-00601-f002:**
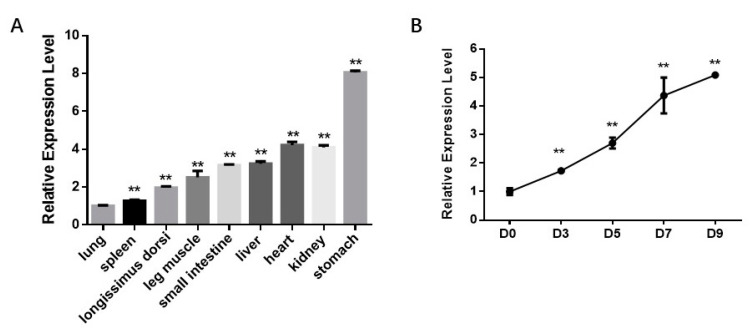
bta-miR-210 expression in fetal tissues and during intramuscular adipogenesis. (**A**) bta-miR-210 expression level in fetal tissues. All the measurements shown are the means ± SEM (standard error of the mean) of three individual biological replicates. (**B**) Relative expression of bta-miR-210 during PDGFRα⁺ cell adipogenic differentiation at Day 0 (D0), Day 3 (D3), Day 5 (D5), Day 7 (D7) and Day 9 (D9), respectively. All the measurements shown are the means ± SEM of four biological replicates from independent cell experiments. ** *p* < 0.01.

**Figure 3 genes-11-00601-f003:**
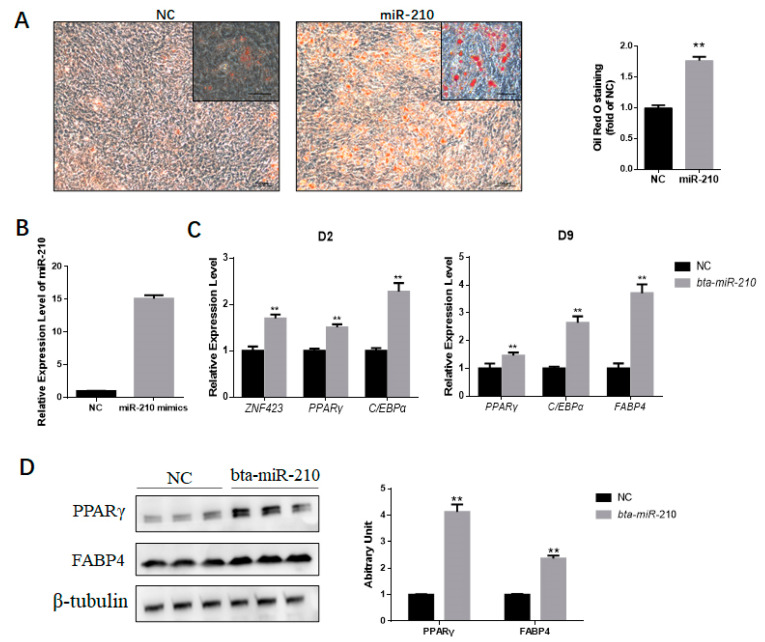
miR-210 enhances the adipogenic differentiation of PDGFRα+ cells. (**A**) The accumulation of lipids was analyzed by the ORO staining of differentiated PDGFRα+ cells transfected with NC or miR-210 mimics. Scale bar = 100 μm, inset scale bar = 50 μm. The quantification of the stained lipid droplets was conducted with eluted ORO stain by detecting the absorbance at 490 nm. (**B**) bta-miR-210 amount after mimic transfection. (**C**) Relative expression of adipocyte-specific genes on Day 2 (D2) and Day 9 (D9). (**D**) Representative Western blot results show PPARγ and FABP4 contents in PDGFRα^+^ cells treated with NC or miR-210 mimics, and the grayscale intensity was quantified. All the measurements shown are the means ± SEM of four biological replicates from independent cell experiments. NC, negative control. ** *p* < 0.01.

**Figure 4 genes-11-00601-f004:**
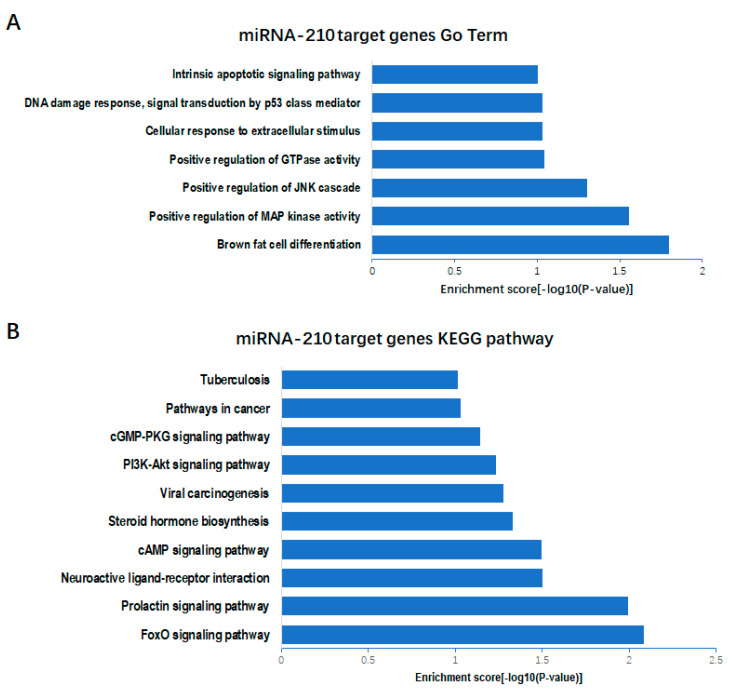
GO(gene ontology) and Kyoto Encyclopedia of Genes and Genomes (KEGG) analysis of miR-210 target genes. (**A**) GO term analysis of the bta-miR-210 target genes. (**B**) KEGG pathway analysis of the bta-miR-210 target genes.

**Figure 5 genes-11-00601-f005:**
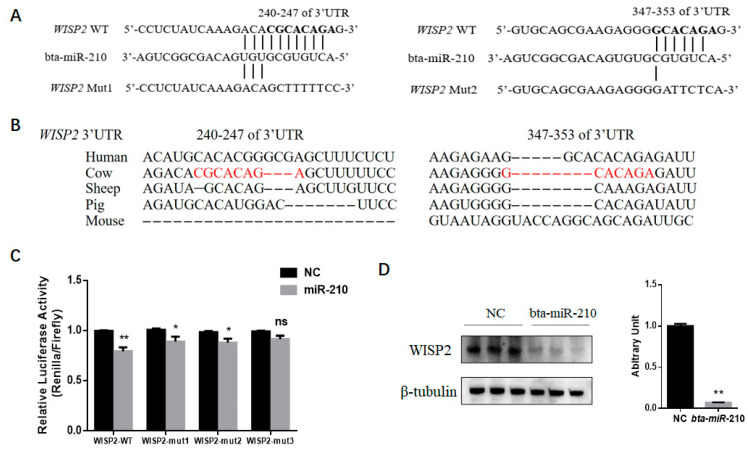
miR-210 regulates the adipogenic differentiation of PDGFRα+ cells by inhibiting *WISP2*. (**A**) Sequence alignments of predicted miR-210 target sites in wild-type and mutant *WISP2* 3′-UTRs. (**B**) Both predicted miR-210 binding sites are not conserved among mammals. (**C**) Luciferase reporter activity of various reporter constructs in cells co-transfected with either miR-210 mimics or NC. *Renilla* luciferase activity was normalized to that of firefly luciferase. (**D**) Representative Western blot results showing WISP2 expression in PDGFRα^+^ cells treated with NC or miR-210 mimics; the grayscale intensity was quantified. All the measurements shown are the means ± SEM of three biological replicates from independent cell experiments. NC, negative control. * *p* < 0.05; ** *p* < 0.01.

**Figure 6 genes-11-00601-f006:**
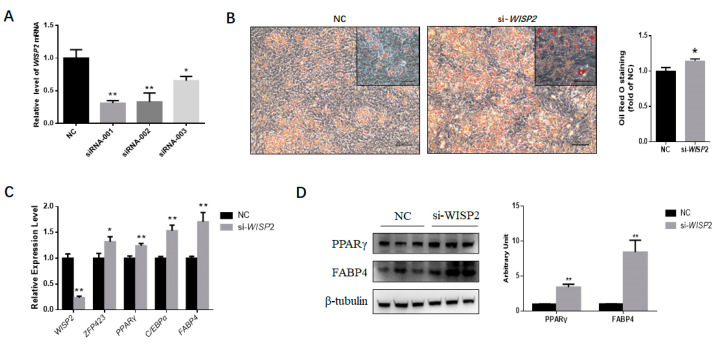
*WISP2* silencing promotes the adipogenic differentiation of PDGFRα+ cells. (**A**) The interference efficiency of siRNAs targeting *WISP2*. (**B**) The accumulation of lipids was analyzed by the Oil Red O staining of differentiated PDGFRα+ cells transfected with NC or siRNAs. Scale bar = 100 μm. The quantification of the stained lipid droplets was performed using the eluted ORO stain via measuring the absorbance. (**C**) Relative expression of adipocyte-specific genes on Day 9. (**D**) Representative Western blot results showing PPARγ and FABP4 expression in PDGFRα+ cells treated with NC or si-*WISP2*; the grayscale intensity was quantified. All the measurements shown are the means ± SEM of four biological replicates from independent cell experiments. NC, negative control. * *p* < 0.05; ** *p* < 0.01.

**Figure 7 genes-11-00601-f007:**
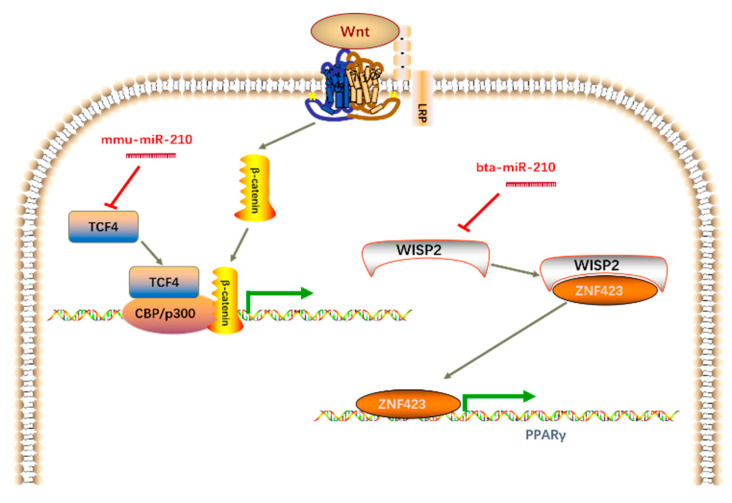
Distinct mechanism of miR-210 in the regulation of mouse and cattle adipogenesis. The canonical Wnt signaling pathway is presented in the figure. Arrows indicate positive regulation, and bars indicate negative regulation.

**Table 1 genes-11-00601-t001:** Sequences of primers used for RT-PCR.

Gene	Primer Sequence (5′-3′)	
	Forward	Reverse
bta-miR-210	ACTGTGCGTGTGACAGCGGC	-
*18S*	CTAACCCGTTGAACCCCATT	CCATCCAATCGGTAGTAGCG
*ZNF423*	GGATTCCTCCGTGACAGCA	TCGTCCTCATTCCTCTCCTCT
*PPARγ*	TGGAGACCGCCCAGGTTTGC	AGCTGGGAGGACTCGGGGTG
*C/EBPα*	TGCGCAAGAGCCGGGACAAG	ACCAGGGAGCTCTCGGGCAG
*FABP4*	GGATGATAAGATGGTGCTGGA	ATCCCTTGGCTTATGCTCTCT
*WISP2*	ACTTGTGTGCCCCTTTGC	AGCGGTTCTGATTGGACACT
*PDGFRα*	TGGCCAGAGACATCATGCA	CTCAGGAGCCATCCACTTCA

**Table 2 genes-11-00601-t002:** Sequence of primers for plasmids construction and mutagenesis.

Name	Primer Sequence (5′-3′)
*WISP2*-Xho I	ATTCTCGAGCCACGTTTCTTCTGAGCC
*WISP2*-Not I	ACTGCGGCCGCAATACGAGTTGGGTTTGATG
*WISP2*-Mut1 Forward	GCCTCTATCAAAGACAGCTTTTTCCAGGAGAT
*WISP2*-Mut1 Reverse	ATCTCCTGGAAAAAGCTGTCTTTGATAGAGGC
*WISP2*-Mut2 Forward	GGTGCAGCGAAGAGGGGATTCTCAGCCTCCTG
*WISP2*-Mut2 Reverse	CAGGAGGCTGAGAATCCCCTCTTCGCTGCACC

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
