# Peer review of "A Novel Mechanism of bta-miR-210 in Bovine Early Intramuscular Adipogenesis"

_genes, 2020, doi:10.3390/genes11060601_

Round 1
Reviewer 1 Report
The manuscript “A novel mechanism of bta-miR-210 in bovine early intramuscular adipogenesis” by Ren et al. analyzes the role of one microRNA, miR-210, in the adipogenic differentiation of PDGFRα⁺ cells collected from bovine fetal skeletal muscle. This adipogenesis would be at the origin of intramuscular fat, whose content is associated with the meat flavor. The authors previously identified miR-210 as overexpressed during bovine intramuscular adipogenesis. However, the molecular mechanism has not been described before. Here, the authors describes in culture the effect of overexpression/downregulation of miR-210 on adipogenic differentiation. They identified one miR-210 target gene, WISP2, as a mediator of the adipogenic differentiation.
This study is well conducted and the results are convincing. However, there are still questions which could be easily answered and could complete this study. In conclusion, I recommend this manuscript for publication in Genes, once the authors will answer the questions raised below.
Remarks:
1- Introduction: Lines 59-61: “WISP2, which could form a cytosolic complex with the transcriptional activator zinc finger protein 423 (ZFP423) preventing ZFP423 from entering the nucleus and activating peroxisome proliferator-activated receptor γ (PPARγ) [22].” This conclusion has not been demonstrated… Do the authors have data suggesting this assessment?
2- In the material and methods section, the composition of the lysis buffer for protein analysis by WB is missing. (line 109)
3- In the figure 1A, 3A and 6B, the Oil red O positive cells and lipid droplets are very difficult to see. A higher magnification, as an inset of the figures, would help.
4- In the legend of figure 1B, the number of replicate should figure, even if it is mentioned in the material and methods section. How many different cultures were performed for each analysis? The standard deviations seem very low. Is it due to the exponential y axis? A control for PDGFRα expression during differentiation of these cells is needed to complete this panel.
5- As miR-210 is expressed in muscles, is miR-210 expressed in myogenic cells? In myofibers? In satellite cells? Is miR-210 involved in myogenic differentiation of myogenic cells? This could be discussed.
6- In figure 2A, the expression of miR-210 is analyzed in several fetal tissues. What is the age of the fetuses? How many embryos were analyzed?
7- In figure 3, the expression of miR-210 should be checked in cells transfected with miR-210 mimics and with mimics NC. How many experiments were performed?
8- In figure 3C, the quantification of FABP4 seems not to be adequate to the blot shown on the side. The differences of expression between treated cells and untreated ones is much less important than shown on the graph.
9- Lines 196-197: It is not clear why the authors chose to analyze this particular target gene, WISP2.
10- Is there any difference in WISP2 protein amount after siRNA transfections?
11- Overexpression of miR-210 enhances overexpression of adipocyte markers. Is it an accelerated process? What about myogenic differentiation? How these results can be interpreted? As WISP2 is an activator of the Wnt pathway, is there any evidence of a activation/repression of the Wnt pathway during adipogenic differentiation of the PDGFRα⁺ cells?
Minor points:
Figure 3C + 5A: misspelling: bat-miR-210 instead of bta-miR-210
Reviewer 2 Report
General:
In general, the study provides an interesting piece of research. Since the description of the use of animals, especially including foetuses, and the data presentation is of insufficient quality, I cannot evaluate the soundness of the study. To me it is not clear why foetuses had to be used for the study.
Introduction and discussion are very short and therefore the article is not easy to interpret for readers from another research area. Therefore the presentation needs to be extensively revised before assessment is possible, again.
Abstract:
- Line 19: please be careful with the word “verified”, in the essence of the word this is barely possible. Furthermore, please state that you showed that in vitro.
Introduction:
- The authors should emphasize which statement is true in general, and which statement refers only to a certain species, e.g. cattle.
- Line 37: “… no fiber number increase…”: to which species does that comment refer? What about tertiary fibers?
- Line 49: “In mice, …”: please indicate if this result was obtained in vivo or in vitro
- Line 50: “Our previous study…”: please summerize the results very briefly for readers not knowing the study.
- Line 52: please explain the term bta
- Line 54: the paragraph is hard to understand without background knowledge (which is only provided later during discussion)
- Line 57: PDGFR – define abbreviation upon first appearance
Materials and Methods:
- Which breed was used? What is the number of animals used?
2.2 cells culture
- Line 71: “between 120 and 150 days…” – how important are the 30 days of difference in age? Do the authors think that they could possibly influence the result?
- How many individuals were used for the study and within each experiment (no n provided)?
- Please describe the method of cell isolation briefly since it is essential to understand the work.
- Where the isolated cells pooled or were foetuses treated separately? Where the cells frozen or directly used for experiments? Please provide the culture conditions (temperature, CO2, …), exact medium composition, the cell density used and specifications for the plates used
- Line 74: “digested” – I don´t think that the authors mean digestion but rather detachment of the cells?
- The authors also seem to use 293T cells – please provide the full name and all information on culturing, etc.!
2.3 adipogenic differentiation
- Line 77: composition of DM?
- Line 81: composition of maintenance medium?
2.4 transfection
- Line 84: “and the sequences are as follows…”: only sequences for WISP are given – what about the other ones?
- Line 86: “… cells were subcultured…” – how exactly?
- Line 89/90: please describe the transfection in more detail.
2.5 qRT-PCR
- Please describe RNA extraction in more detail. Using trizol is surely not sufficient.
- Line 97: “The primers for miR-210…”: Do the authors mean bta-miR-210?
2.7 Western blot
- How where the protein samples obtained? Was the protein amount determined? Please provide the exact composition of the lysis buffer. How much protein was loaded onto the gel?
- Please provide time and temperature for the incubation steps.
- Where the secondary antibodies conjugated?
- Where two proteins (target and reference) stained on the same membrane or was it stripped?
- Bioinformatics Analysis
- Please describe the analyses in more detail.
2.9 DNA constructs and luciferase
- Please describe the luciferase assay in more detail, e.g. luciferases, promotors, …
2.10 statistical analysis
- The authors only seem to describe statistics for qPCR – what about the other experiments?
- Did the authors check for primer efficiency, RIN and the stability of the used housekeeping gene?
- The authors mention technical replicates – did they use biological replicates? Please provide n for every experiment and if accounts for biological replicates.
- The authors used t-test: did they test for normality and equal variance beforehand?
Figures:
- Please provide the number of biological (!) replicates for each figure.
- What do the authors mean by “independent experiments”?
- In general, the normalization within graphs is misleading and incorrect and therefore not acceptable – in every graph only one value should be used for normalization. E.g. Fig. 1B gives the impression that the expression of all three genes at d0 was the same – which is certainly not true. This has to be revised for every figure!
- Fig. 2: statistical analysis is missing.
- Fig. 2A: It was not stated in methods that tissues other than LD muscle were taken from the animals.
- Fig. 4: hard to read because of quality
- Fig. 5, line 219: “at least” – specify the number
- Fig. 6B: scale bar cannot be seen
- Fig. 7: Why mouse and cattle?
Discussion:
- Line 256: “…especially in the early developmental stage” – why did the authors decide to use that stage? What are the advantages? Would postnatal analyses be possible as well? Especially considering ethical aspects about using foetuses.
- Line 258: “in farm animals” – which species?
- Line 261-282: this is a nice presentation of facts, but barely a discussion.
- Line 291: “… in bovine adipogenesis…”: the figure indicates also mouse.
Ackknowledgements:
- The authors forgot that paragraph.
Round 2
Reviewer 1 Report
The authors properly responded to all the comments I had on the manuscript.
Author Response
Thank you for all of the suggestions which help us make our manuscript better.
Reviewer 2 Report
General:
The authors improved the manuscript according to some of the reviewers questions. I would like to recommend the authors to be more accurate with their methodological descriptions in this manuscript and in the future, although this will certainly take some time.
I have still some concerns, which have to be answered. Although they are minor, this is in my point of view a prerequisite to consider the manuscript for publishing.
Abstract:
- Line 20: in vitro – write in italics
Introduction:
- Line 37: “… no fiber number increase…”: The authors added the species. Please be aware that the fact is not true for every species.
- Line 50: “Our previous study…”: The authors now summarize the results. Please provide the used model in addition.
- Line 57: PDGFR – define abbreviation upon first appearance. (I noticed that the authors explain the abbreviation in the abstract, but it would be preferable to explain it upon first appearance in the main text as well, as it was done for e.g. IFM.)
- Line 57: spelling error technique
Materials and Methods:
- The authors now state that they used 3 foetuses for the whole study. Therefore, these foetuses seem to be the biological replicates (performing an assay several times with cells from the very same foetus is not a biological replicate!). Consequently, in my point of view it is not possible for the authors to obtain more than 3 biological replicates – although this is stated in some figure legends. I am afraid that the authors mixed up technical and biological replicates. This has to be clarified. I understand that it might be difficult to obtain enough material from animals for biological replicates, sometimes. But still, this has to be stated correctly or mentioned as a limitation of the study. Especially considering that foetuses used where not exactly the same age.
2.2 cells culture
- Please provide exact information for the medium used. “DMEM” is certainly not specific enough. Did the authors really use no antibiotics in some media?
- Line 76: “between 120 and 150 days…” – The authors answered my question regarding this point and I would like to suggest to add this into the manuscript.
- Specification for (12well) plates and CO2 for cultivation is still missing.
- Source of HEK cells and some culturing conditions are still missing.
2.3 adipogenic differentiation
- Line 101: composition of DM? – Did the authors really add no FBS?
2.4 transfection
- Line 108: Details on transfection are still missing. E.g., did you use Lipofectamine? Did you follow a published protocol? Did you use a special medium for transfection?
2.5 qRT-PCR
- Information of RNA extraction is still missing. Did you use a kit?
2.7 Western blot
- The authors state that each protein was stained independently. Does that mean on different gels/membranes or was the membrane stripped to stain for the reference protein?
2.9 DNA constructs and luciferase
- Please describe the luciferase assay in more detail, e.g. luciferases, promotors, … - Are the authors sure that they didn´t mix up renilla and firefly luciferase? Which promotor was used for control luciferase? TK?
2.10 statistical analysis
- The description is still not sufficient.
- Information on equal variance test is still missing.
Figures:
- Please see my initial comment regarding biological and technical replicates.
- In general, the normalization within graphs is misleading and incorrect and therefore not acceptable – in every graph only one value should be used for normalization.
I cannot accept the answer of the authors regarding this issue since the current presentation is misleading! Even if the authors do not compare the values of the different controls, the reader should be able to do so. Change the graphs! If you would like to keep the normalization, every normalized value and the according bars have to appear in a separate diagram.
- Fig. 2: statistical analysis was not conducted, please explain why.
- Fig. 7: Explanation of figure is missing.
Discussion:
- Line 290 + 299: sentence not complete
- Lines 330-334: Are there also implications for other species? Although the authors work on cattle in this study, it might also be important for other species.
Author Response
Dear reviewer,
Thank you for your comments and suggestions. We have revised the munuscript according to your kind suggestions. We hope revised manuscript could be acceptable for publishing.
Sincerely!
Lupei Zhang
